# Bacteriophages of the Order *Crassvirales*: What Do We Currently Know about This Keystone Component of the Human Gut Virome?

**DOI:** 10.3390/biom13040584

**Published:** 2023-03-24

**Authors:** Linda Smith, Ekaterina Goldobina, Bianca Govi, Andrey N. Shkoporov

**Affiliations:** APC Microbiome Ireland, Department of Medicine & School of Microbiology, University College Cork, T12 YT20 Cork, Ireland

**Keywords:** *Crassvirales*, virome, human gut microbiome, metagenomics

## Abstract

The order *Crassvirales* comprises dsDNA bacteriophages infecting bacteria in the phylum Bacteroidetes that are found in a variety of environments but are especially prevalent in the mammalian gut. This review summarises available information on the genomics, diversity, taxonomy, and ecology of this largely uncultured viral taxon. With experimental data available from a handful of cultured representatives, the review highlights key properties of virion morphology, infection, gene expression and replication processes, and phage-host dynamics.

## 1. Introduction

The crAss-like phages are currently thought to be the most abundant double-stranded DNA bacteriophages in the human gut. Yet, despite a wealth of metagenomic data available from the human gut, this highly abundant viral group escaped discovery until 2014 [1]. The near accidental discovery of crAssphage (later dubbed p-crAssphage, or prototypical crAssphage), a bacterial DNA virus (phage) that shared no DNA and very little protein homology with any of the known viruses, highlighted the very limited capacity that existed at the time to discover completely novel viruses, including very prominent ones. The discovery of p-crAssphage has invigorated microbiome researchers to examine the virome portion of the human gut and other environments more closely [2,3,4,5].

Once advanced tools for detection of putative viral genomes became available, a more rigorous examination of previously untraceable “viral dark matter” in metagenomic data revealed a whole plethora of novel viral clades, including “Flandersviridae”, “Quimbyviridae”, Gubaphage, and giant Lak phages [6,7,8,9]. The p-crAssphage itself turned out to be not an orphan but rather a member of an expansive and highly abundant clade of intestinal and environmental phages, initially known as crAss-like phages, now forming the order *Crassvirales*. While p-crAssphage and related phage strains (members of the recently created new species *Carjivirus communis* in the family *Intestiviridae*, subfamily *Crudevirinae*) are the most prominent members in the gut microbiomes of westernised populations [10,11], other genera, subfamilies, and families of *Crassvirales* appear to be dominant in human populations adhering to a pre-industrial lifestyle and diet and in non-human primates [10,11,12].

Current analyses estimate crAss-like phages to be present in 73% of global human gut metagenomes, where they can comprise up to 99% of the viral metagenomic reads [10,13]. Found even in remote populations, it is a ubiquitous global virus clade, and its detection in Old and New World primates suggests a possible ancient coevolutionary relationship with humans. Established crAssphage populations are highly stable in the human gut [10,13] with the same viral strain having been demonstrated to persist up to 4 years after original sampling [13,14]. As no detrimental associations between human health and the presence of crAssphage have yet been found, it has been deemed a benign entity in the gut microbiome [11,12]. A beneficial role in the human gut has been suggested by observations of depleted crAss-like phage populations in patients with inflammatory bowel disease, and a destabilisation of p-crAssphage is linked to obesity and metabolic syndrome [15,16].

Since their detection, the crAss-like phages have drawn great interest, yet their role in the gut microbial ecology remains obscure. Their known bacterial hosts, members of the phylum Bacteroidota, mainly family *Bacteroidaceae*, are essential mutualistic bacteria of the human gut, present at high densities, and are well documented to perform various functions beneficial to the human host, such as immunomodulation, colonisation resistance against invading pathogens, biosynthesis of vitamins, and cooperation with other commensal and mutualistic microbes through cross-feeding networks [17,18,19,20,21,22,23]. Phages can manipulate bacterial populations and communities through a variety of mechanisms: by lysing their bacterial targets (lytic infection) and affecting community structure and cross-feeding networks [24], by integrating prophages into the host genome (lysogeny) and altering host metabolism and fitness [25], and by transferring genes (phage transduction), which can confer advantages to the host (e.g., antibiotic resistance [26,27]). CrAss-like phages seem to be strictly lytic and incapable of lysogeny [28]. At the same time, longitudinal observations of the human gut microbiome, phage-host colonisation trials in germ-free animals, and in vitro phage-host co-cultures demonstrated that crAss-like phages can efficiently co-exist with their host bacteria without inflicting excessive damage at the population level [13,20,21,22,29].

What is the ecological role of this highly abundant virus infecting the family *Bacteroidaceae*, and what is the secret of their ecological success? Are they crucial for maintaining the stability, diversity, and functional versatility of their host populations, or are they just selfish parasites whose presence their bacterial partners have no choice but to tolerate? A macro view of the gut virome at the population and community levels would suggest there is either an ongoing active arms race between crAss-like phages and their hosts or an “uneasy truce” between the two in which some form of mutualistic interaction or collaboration between these entities has been established [30]. What aspects of their biology have enabled them to become extremely successful colonisers of the human gut? Several crAss-like representatives have been isolated in culture and have been the subjects of experiments designed to address these questions [18,19,20,21,22,31].

The discovery of crAss-like phages is exemplary in proving we have only begun to scratch the surface in uncovering the global viral diversity. Unique in the sense that it was discovered as an orphan genome before its isolation in the lab, crAssphage set a trend for virus discovery in silico, whereas historically phages would be isolated in the lab before their characterisation [32]. The ongoing metagenomics and metatranscriptomic revolution [33], facilitated by access to scalable computing resources and growing databases, is fuelling the pace of viral discovery in global human and mammalian gut samples as well as in any possible sampled environment [6,7,8,9,34,35].

In this review, we attempt to briefly summarise available data on the diversity, taxonomy, and ecology of cultured and uncultured *Crassvirales* species, as well as to briefly review the history of their discovery. We will also highlight recent progress in understanding their virion molecular structure, infection mechanisms, and phage-host dynamics, achieved with the help of a limited number of cultured species of *Crassvirales*.

## 2. Metagenomics, Diversity, and Ecology of *Crassvirales*

p-crAssphage (*Carjivirus communis*) was discovered from metagenomic assemblies of pooled, short sequencing reads sourced from faecal samples of four pairs of monozygotic twins and their mothers [1]. Named in homage to the cross-assembly (crAss) algorithm used to generate its apparently circular, complete genome of 97 kb, p-crAssphage represents an “average” of perhaps several closely related strains present in the gut of those subjects. This was the first report of a supposed unique orphan phage with unusually high prevalence in the human gut microbiome—the majority of people from Western cultures tested positive, with levels sometimes reaching >90% of the total virome and >20% of the total faecal DNA. Although the functions of most of its genes, its potential morphology, and its bacterial host remained unknown, nucleotide similarities to CRISPR spacers in bacterial genomes pointed to a possible host in the genus *Bacteroides* [36]. A global phylogeographic study by Edwards et al. employed targeted sequencing of three PCR-amplified p-crAssphage genomic regions from wastewater treatment facilities around the world, as well as mining of 95,552 metagenomic datasets [11]. This effort resulted in 32,273 different p-crAssphage sequences from 67 countries across six continents and proved the ubiquitous presence of this phage in the human gut around the world. Analysis of the metadata in the Life-Lines-DEEP cohort did not reveal any significant association between any disease and health parameters and the p-crAssphage levels [11]. However, weak correlations with dietary factors (protein, carbohydrates, and calorie intake) were observed. Another biogeographic study of p-crAssphage observed that industrialisation plays a role in the dominance of p-crAssphage in the human gut [12]. Samples from Hadza individuals in Tanzania were crAssphage-negative (n = 67), while only 2/25 samples acquired from Matses Peruvians were crAssphage-positive. These populations are known to subsist on a primarily plant-based hunter–gatherer lifestyle. Similarly, p-crAssphage was found in low abundance in remote populations in rural Malawi and the Amazonas of Venezuela [11]. In contrast to these rural populations, prevalence among urban individuals was high: —35.7% of U.S. residents and 14% of Chinese residents [12]. These findings agree with *Bacteroides* also being associated with industrialised human populations and diets [37].

A study of longitudinal virome colonisation following FMT from healthy donors to patients with recurring *Clostridium difficile* infection (rCDI) showed crAssphage can effectively populate and persist long-term in the recipient gut. Given that rCDI patients began with a significantly lower crAssphage population than healthy donors, they quickly scaled up to an equivalent level to that of the healthy donors and remained stable for over 1 year. The crAssphages are likely an essential component of a healthy gut ecosystem and provide an important dimension in community maintenance and stabilisation [38]. Indeed, whole-virome analysis of inflammatory bowel disease (IBD) and ulcerative colitis (UC) datasets shows IBD-specific alterations in the gut virome, specifically a loss of crAss-like phages and *Microviridae*, identifying crAss-like phages as critical components of the core virome in healthy individuals [16]. Another study reporting loss of phageome stability in disease found reduced abundance and diversity in children with obesity and obesity-related metabolic syndrome [15]. The fact that a virulent crAssphage core represents a healthy microbiome at equilibrium and that virome dysbiosis can be overcome by FMT highlights the importance of considering the viral portion of a patient’s metagenome when considering FMT therapy [38].

Further work made it apparent that p-crAssphage was not an orphan species but rather represented a whole family of cryptic bacteriophages termed “crAss-like phages”. In 2018, Yutin et al. pioneered the detection of crAss-like phages in diverse environments by applying advanced approaches to detect distant protein homologies and genome synteny on candidate sequences extracted from expanded NCBI nucleotide and whole-genome sequence (wgs) databases [39]. Multiple clades of phages with highly divergent but clearly colinear and evolutionary related to p-crAssphage genomes were detected populating such diverse environments as the human microbiome (IAS virus), marine sediment (*Cellulophaga* phage phi14:2), the *Populus* rhizosphere (*Chitinophaga* sp. YR573), the termite gut (*Azobacteroides* phage ProJPt-Bp), and a fish pathogen (*Flavobacterium* phage Fpv3) [39]. The extent of crAss-like diversity in the human gut was further explored by Guerin et al., who clustered 249 unique crAss-like phage genomes, assembled from 702 human faecal virome samples, based on sharing of orthologous genes, and identified ten candidate genera (subfamilies in the present classification) and four candidate subfamilies (families at present) [10]. The frequency of detection of different crAss-like phage genera was significantly associated with country of residence, health status, and the age of the donor, with the lowest counts seen in Irish and Malawian infants, as well as IBD patients in the U.S. As well as having a lower relative abundance, the genera of crAss-like phages detected in rural Malawi separated from the Western healthy and diseased cohorts, showing genus I (subfamily *Crudevirinae*, family *Intestiviridae*) as the most prevalent in Western populations regardless of age. By contrast, Malawian infants primarily carried candidate genera VI, VIII, and IX (*Steigviridae*, *Obtuvirinae*, and *Oafivirinae*).

CrAss-like phages are not only restricted to the human gut but have also been found in other mammalian faeces, such as those of rhesus macaque, pig and cat [34,40]. Edwards et al. demonstrated the presence of candidate genera I, III, and IX in apes and Old-World and New-World monkeys [11]. Given the narrow focus of many studies on a single species (p-crAssphage), much less is known about the global phylogeography of the other clades of crAss-like phages. If shotgun metagenomics/metaviromics data from each of these sources were to be extensively analysed, a more granular and comprehensive picture of the distribution of crAss-like phages and the biogeographic mapping of more distantly related members of this viral group would likely emerge.

In addition to the discovery of diverse crAss-like phages, distant protein homology detection improved their genome annotation, proposed a possible morphology, and identified specific elements of their lifecycle. Due to tail protein homology between crAss-like phages and bacteriophage P22, a known short-tailed podovirus, it was predicted that crAss-like phages would similarly possess short, stubby tails [39,41]. Transmission electron microscopy examination of crAss-like-rich faecal filtrates confirmed a podovirus morphology, with 82% of virus particles having a short tail form [10]. Insights into their lifecycle were gleamed from the discovery of an unusually large RNA polymerase gene (up to ~6000 aa), showing that crAss-like phages encode their own active virion-associated RNA polymerase (vRNAP), which they pre-package in the capsid during assembly [10,18]. The presence of two conserved motifs of similarity to the β′-subunit of the bacterial RNAP suggests that it may have originated from the fusion of two bacterial RNAP subunits, becoming a β-β′ RNAP fusion [39]. These giant RNAPs have become a hallmark of crAss-like phages. Other identified proteins implicated in replication were two families of A and B DNA polymerases (DNAP) [39]. Of the replicative units, DNA primase is the only gene to be universally conserved across the crAss-like family. Moreover, crAss-like phage primases form a monophyletic clade and are homologous to Bacteroidetes primases, suggesting ancient evolutionary acquisition of Bacteroidetes DNA primases by the earliest crAss-like phages [39]. CrAss-like genomes also contain carbohydrate-binding domains (BACON domains), which are highly homologous to a protein found in *Bacteroides* [1]. These gene capture events provide an insight into the long-standing associations between Bacteroidetes and the crAss-like phages.

## 3. Co-Evolution with Bacterial Hosts

Antagonistic interactions between phages and their hosts drive their co-evolution, encouraging both parties to develop novel strategies to increase their fitness and overcome any defence mechanisms developed by the opposite party [30,42]. Gene duplication is one such strategy found in crAss-like sequences that is speculated to bestow replicative advantages. For instance, p-crAssphage has two copies of *repL*, required for the initiation of DNA replication, which arose from an ancient gene duplication event [43]. CrAss-like phages assigned to candidate genus I (*Crudevirinae* subfamily of *Intestiviridae*) have been found to encode diversity-generating retroelements associated with tail protein genes (DGR) [44]. DGRs employ reverse transcriptase to selectively mutate a stretch of DNA coding sequence, giving rise to the creation of original hypervariable tail proteins [45]. This built-in DNA mutator cassette is a versatile protection mechanism that potentially enables phages to counteract bacterial defences (resistance) or exploit alternative bacterial hosts by altering their tail proteins involved in host recognition.

An analysis of nucleotide polymorphisms and the rate of non-synonymous to synonymous changes (dN/dS) in crAssphage genomes from a mother-child cohort with low interpersonal viral diversity showed rapid evolution and strong positive selection for mutations in genes coding for RNAP and tail proteins [46]. Considering the crucial role tail proteins have in bacterial host recognition and the overcoming of acquired phage resistance, it is fitting that their genes would be targets of active selection. One specific target of elevated variation is the BACON domain-containing protein, which is located proximal to the tail protein complex and is thought to have a role in binding to the cell surface of *Bacteroides* [47]. The rapid evolution of these genes could reflect crAss-like phage’s response to a Red Queen scenario where they are adapting to the host bacteria continuously cycling their cell-surface receptors.

## 4. Proposed Taxonomy of *Crassvirales*

A 2021 taxonomic proposal to the International Committee for Taxonomy of Viruses called for the creation of a new order, *Crassvirales*, encompassing four new families (*Intestiviridae*, *Crevaviridae*, *Suoliviridae*, and *Steigviridae*), ten new subfamilies, 42 new genera, and a total of 73 new species [48]. The percentage of shared orthologous genes (proteins), as well as single gene phylogenies reconstructed based on amino acid sequences of the large terminase subunit (TerL), the major capsid protein (MCP), and DNA primase, were used to demarcate families, subfamilies, and genera (Figure 1). A phylogenetic comparison of both MCP and TerL proteins places each of the four families on separate branches, with branch members sharing a minimum of 17% of orthologs. Monophyletic groups clearly emerging from the four family clades were deemed sub-family taxa. Clustering of subfamily genomes based on gene sharing showed 27–79% of proteins within each subfamily. A total of ten subfamilies were proposed, which roughly correspond with formerly proposed candidate genera (I-X), with the exception of former candidate genus VI, which was reformed into a single family, *Steigviridae*, with no clearly definable subfamilies in it. Species are defined as sharing 95% sequence identity over 85% of the complete genome length, in accordance with the advised MIUViG standard [49]. Only four species listed in the latest taxonomic proposal have been cultured [18,19,20,31], with the remaining majority originating from metagenomic assemblies (circular contigs). Synonyms of the word “crass” were affixed to create sub-family names, while the names of the four main families are language derivatives of the word “intestines”: *Intestiviridae* (Latin), *Crevaviridae* (*creva*, Bosnian), *Suoliviridae* (*suolet*, Finnish), and *Steigviridae* (*stéig*, Irish).

## 5. Hallmark Genetic Features

An analysis of 596 diverse circular crAss-like phage genomes uncovered some unusual aspects of crAss-like phage architecture and biology [8]. In line with previous host assignments, CRISPR spacer searches of the prokaryotic genome database assigned the majority of crAss-like sequences to hosts in the phylum Bacteroidetes. Lower frequencies of spacer matches aligned to other bacterial phyla such as gut Firmicutes and Proteobacteria, and a minority proportion aligned to Archaeal genomes. The study also consolidated the genomic trademark of crAss-like phages as a syntenous block of eight highly conserved genes encoding the structural proteins MCP, TerL, portal protein, integration host factor (IHF), tail stabilization protein, tail tubular protein, and two uncharacterized proteins. Yutin et al. (2021) also identified two groups of environmental and gut phages appearing related to the core crAss-like phage phylogenetic assemblage, which were named as groups Zeta and Epsilon (continuing the original system of naming *Crassvirales* families as Alpha, Beta, Gamma, and Delta) [10]. These groups were characterised by larger genome sizes, ranging from 145 to 192 kb, almost twice as large as p-crAssphage’s 97 kb.

An interesting observation was the alternative choice of family A or B DNA polymerases by different clades. Families *Intestiviridae*, *Crevaviridae*, and *Suoliviridae* (Alpha, Gamma, and Delta groups) exclusively encode PolB, while Zeta and Beta groups (the latter being family *Steigviridae*) have no preference and can either carry PolA or PolB. When co-occurring within groups, PolA and PolB occupy the same position in the genome and are thought to be swapped in situ by homologous recombination. It is hypothesised that this phenomenon has arisen due to the selective pressure presented by an undefined *Bacteroides* defence mechanism. In addition to DNA polymerase switching, another notable finding was the use of alternative genetic codes, specifically in 243 crAss-like phages in the Beta, Delta (*Steigviridae, Suoliviridae*), and Zeta groups that reassign TAG stop codons as glutamine or TGA stop codons as tryptophan. Phages in the Zeta groups can switch genetic code at short phylogenetic distances, with closely related branches interspersed with phages reassigning TAG or TGA or using the standard code. An established method of phage codon reassignment is through the use of suppressor tRNAs, which encode an anticodon complementary to the standard stop codons charged with an amino acid [8]. Some crAss-like phages, especially in the Beta (*Steigviridae*) and Zeta groups, encode a plethora of tRNAs (1–34), with one Zeta phage (NCBI accession OJOH01000017) encoding 34 tRNAs. The encoding of suppressor tRNAs is not completely congruent with stop codon reassignment [8,50]. For example, in some Zeta clades, despite encoding suppressor tRNAs, no reassignment occurs, and in other cases where reassignment occurs, no suppressor tRNAs are present. This leaves the mechanism and motive behind genetic code switching in crAss-like phages ambiguous.

Another distinct feature of crAss-like phages that has not been observed in other phages is the presence of at least one nuclease of the PDDEXK family, with many encoding two or three of these. The exact function of these nucleases is unclear, but they are presumed to have a role in reproduction, with some phages having one variety located in the structural block (PDDEXK_a), another in the replication block (PDDEXK_b), or yet another inside a large protein associated with transcription. The blocks of short, unknown genes common in many crAss-like genomes that share no homologs outside the crAss-like cohort are probably the most intriguing. Postulated to be genes in their defence armoury, specifically anti-CRISPR proteins (Acr), their purpose is still up for debate as a machine learning Acr model failed to confirm their function.

Introns, a group of mobile genetic elements that act as self-splicing ribozymes (an RNA with enzymatic activity) [51], and inteins, proteinaceous parasites that perform self-splicing by cleaving peptide bonds [52], are typical of the genomic landscape in many bacteriophages [51,53]. The crAss-like phages of the Delta (*Suoliviridae*) and especially Zeta groups have a remarkable density of introns and inteins. In Delta phages, group I introns are inserted in the core phage genes—MCP, PolB, and primase—with TerL containing an intein. A stark example of this intron infestation is found in a Zeta phage’s (NCBI accession number: OHFV01000001) MCP gene, which contains three Group I self-splicing introns within its ORF fragments. In the Zeta group, their predicted RNAP ORFs are drastically fragmented by numerous stop codons and frame shifts, some occurring within the highly conserved DxDxD motif of RNAPB’. The disrupted nature of the genes highlights the complexity and nuance of annotating crAss-like phage genomes. Taking all of this into consideration, crAss-like phages likely employ unconventional encoding and expression mechanisms that have yet to be understood.

## 6. Alternative Genetic Coding Strategy: Experimentally Confirmed

LC-MS/MS metaproteomics of two metagenomic samples containing two different crAss-like phage genomes (one *Suoliviridae* and one Zeta group) identified phage peptides from MS/MS spectra. When mapped to the optimal gene predictions using both the standard genetic code (translation table 11) and table 15 (suppression of TAG stop codon with glutamine), around half of the peptides were found to only map to structural proteins (capsid, portal, and tail-associated) predicted using table 15. MS/MS peptide spectra supported the reassignment of the TAG codon to code for glutamine (Q). The contrast in mapping efficiency and correctness of annotation between the standard and non-standard genetic codes is stark and shows that consideration should be taken when predicting crAss-like phage genes [54].

Stop-codon recoding is widespread among different phage clades in the human gut microbiome, with crAss-like phages and Lak phages being found to recode the TAG or TGA stop codon (genetic codes 15 and 4) [8,55]. CrAss-like phages using an alternate code infect standard-code hosts (genetic code 11) [8]. It is uncertain why some phages have evolved genetic codes incompatible with host translation systems. Interestingly, only 35%/40% of TGA/TAG-recoded genomes contain a corresponding tRNA suppressor. Do the other 60% rely on non-standard code used by the host, or do they use an unknown mechanism to accommodate their alternate code? Importantly, CrAss-like phages, for the most part, use alternative coding for their “late” structural and lysis genes (which constitute a single cohesive transcriptional unit). Therefore, the use of recoded stop codons might be required for regulating the precise timing of structural protein expression and cell lysis. Specific gene families seem to be enriched with recoded stop codons: lysozyme-type amidase and spanin/holin (both proteins involved in the lysis-lysis cassette). In such scenarios, regulated genetic code switching might be helping to prevent premature lysis (abortive infection activated by some anti-phage defence systems). Recoding has also been suggested to enable phages to sense co-resident phages and coordinate their lysis timing. Temperate gut phages (non-crAss-like) also employ genetic code switching to control lysis-lysogeny decisions. Given that crAss-like phages are not thought to be lysogenic, it is still possible that recoding contributes to the regulation of lysis (productive vs. abortive infection) in order to fine-tune the steady state in phage-host pairs and ensure long-term persistence [50].

## 7. P-crAssphage as a Marker for Faecal Contamination

Given p-crAssphage’s ubiquity in human faecal samples, an industrial application of its prevalence has been its utilization as a microbial source tracker (MST) marker of human faecal contamination and pollution in environmental water, surfaces, and wastewater treatment [56,57]. One interest in using crAssphage as a marker for faecal pollution in environmental waters was borne out of the ability to distinguish human faecal waste from other animal faecal pollution [57]. That withstanding, crAss-like variants of animal gut origin are being researched to reliably assess animal faecal pollution of wastewater, with only 61% of wastewater confirmed positive for animal faecal matter showing positive for crAssphage [58].

## 8. Identification of Bacterial Hosts for crAss-like Phages

The classic approach to the isolation and identification of bacteriophages in environmental samples and linking them to their bacterial hosts is based on plaque assays. Subsequently, this bacteriophage can be further propagated from the plaque and studied. The success of the plaque assay approach in searching for phage-host pairs depends on several factors: the choice of the right bacterial host strain, the ability of the bacteriophage to form visible plaques, the suitability of the medium and other growth conditions for the bacterial growth and plaque formation, and the amount of infective phage particles in a lysate obtained from a sample. A successful combination of all these factors that would yield a plaque is not always possible, especially for such complex communities as the gut microbiota, which contains predominantly temperate phages [59,60,61] and phages infecting bacteria that remain mostly uncultured under laboratory conditions [1]. Despite all its weaknesses, the traditional plaque/spot assay approach was used for the isolation of several crAss-like phages, such as phi14:2, crAss001, DAC15, and DAC17 [19,31]. The first member of the now-extinct *Crassvirales* to be successfully isolated by this traditional method was in fact the *Cellulophaga* phage phi14:2. This marine virus was isolated and classified as a member of the family Podoviridae the year before the discovery of p-crAssphage and the sudden rise of interest in crAss-like phages as a novel taxonomic entity [1,31]. Phage enrichment in a liquid culture is a culture-based technique that does not rely on plaque formation. This technique, in combination with next-generation sequencing of the enriched virus-like particle (VLP)-associated DNA fraction, has been used for human gut phageome studies and resulted in the discovery of a significant amount of novel phage-host pairs, including those for crAss-like phages [23]. This plaque-independent, culture-assisted metagenomics approach was used for the identification and characterisation of crAss-like phages ΦcrAss001 [18] and ΦcrAss002 [20]. Additionally, the first crAss-like phages replicating in *Bacteroides uniformis* cells were detected with similar methods [21].

The paucity of cultured crAss-like phages has prompted the development of alternative approaches for identification of the phage-host pairs. The *proximity ligation* approach is based on chemically cross-linking DNA fragments that are in close proximity to each other in three-dimensional space. Cross-linking of the fragments results in them belonging to the same contigs after sequencing and data processing. This method allowed the assignment of several members of the crAss-like family to their bacterial hosts. A range of different hosts, all belonging to the Bacteroidetes phylum, was identified for 17 crAss-like phages using meta-HiC.

*Microbe-seq*, a new complex method of single-cell sequencing, is one of such approaches that uses single amplified genomes (SAG) [62]. This is a culture-independent method that allows for genomic data with strain resolution and is also capable of detecting host-phage interactions within complex populations. As phages associated with bacteria are encapsulated in the same single-cell droplets, the bacterial and viral genomes are linked in the output data by droplet-specific tags. It is thus possible to search for these phages in the resulting genomic data and track their connection with a specific bacterial strain. Using this approach, *Bacteroides vulgatus* from human gut samples was found to be significantly associated with the p-crAssphage and was therefore predicted to be its host. This result is consistent with other studies that propose the host of the crAssphage to be from the *Bacteroides* genus [63,64].

CRISPR spacers analysis resolved the crAss-like phage hosts at the order level (*Bacteroidales*), while more precise host identification by this method is problematic due to horizontal gene transfer leading to distant bacterial species sharing the same CRISPR direct repeats, which hinders taxonomic assignment of CRISPR regions themselves [64]. The correlation between bacterial and viral relative abundance pointed towards *Bacteroides vulgatus* and *Ruminococcus* spp. as possible hosts for crAss-like phages [65]. P-crAssphage was found to strongly co-occur and correlate quantitatively with *Bacteroides dorei* [66]. The credibility of this prediction remains questionable, as the results of the microbe-seq laboratory approach do not support it [62]. Other approaches that can potentially be applied to identify hosts for diverse uncultured crAss-like phages are listed in Box 1.

Box 1Methods of host assignment for uncultured (difficult-to-culture) bacteriophages. Many of the viruses discovered with metagenomics approaches have never been isolated in culture. The only methods that allow to link these uncultured phages to the bacterial hosts are bioinformatic predictions. The coevolution of phages and their hosts leaves specific signatures in the genomes of both. The signals for host-phage identification could be abundance profiles, codon usage similarity, genetic similarity, and CRISPR spacer matches [36,64]. Examples of the tools are HoPhage, HostFinder, PHIAF, and VirHostMatcher. Existing bioinformatics tools for in silico predictions of phage-host interactions were reviewed elsewhere [67]. One of the most widespread experimental, culture-independent methods of identifying and describing phage-host interactions is proximity ligation [68,69,70,71]. Recently, an improved protocol based on proximity ligation, called meta3C, was developed. Another recent modification of the proximity ligation method is the cross-linking of specifically bacterial ribosomal RNAs to the transcripts. rRNA-mRNA chimeras where mRNA belongs to a virus allow to identify the host of this virus—the organism that translates its transcripts. Approaches that utilize single amplified genomes (SAG) are another powerful culture-independent tool for linking phages to hosts. In these approaches, DNA from individual cells is isolated and sequenced. The first step is separating individual cells from each other, which can be conducted with microfluidics or with flow-cytometry techniques. In the next step, the cells are lysed, their genomes are amplified, and they are subjected to next-generation sequencing. Applied to phage-bacteria interactions studies, SAG approaches reveal associations between viral DNA and bacterial DNA of different species and strains. These associations serve as the basis for predictions about which bacterial species serve as hosts for specific phages [72,73,74,75,76]. A virus adsorption-based approach to link bacteriophages from environmental samples to their bacterial hosts is adsorption sequencing (AdsorpSeq). It is based on the interactions between bacteriophages and receptors on the surface of their hosts. The phages that were able to bind to the cell envelopes isolated from a putative host are separated from free (unbound) phages with agarose electrophoresis and then identified by sequencing. This method was verified for several distinct species of Gram-negative bacteria, while its suitability for Gram-positive bacteria remains to be investigated [77]. One of the main drawbacks of all host identification methods that are based on the adsorption of phages to their bacterial hosts is the fact that not all phages that are able to bind to a bacterial surface are able to infect this specific bacterium. Therefore, the range of phages identified for a set of hosts with one of these methods is likely to be wider than the range of phages that successfully infect these hosts. Viral tagging is another method based on phage adsorption. In this approach, viruses from environmental samples are labelled with fluorescent dye and mixed with supposed bacterial hosts [78,79]. Then the cells are separated by flow cytometry and activated by fluorescence. Infected cells are collected for viral DNA extraction and amplification that yields viral-tagged metagenomes. The single-cell viral tagging was implemented to understand the network of phages and their hosts in the human gut; the study yielded hundreds of predicted phage-host pairs. Droplet digital PCR (ddPCR) [80,81,82] allows for the identification of the co-localisation of viruses and bacterial cells. Cells from an environmental sample are diluted and directly embedded in droplets or in microfluidic PCR chambers in such a manner that most droplets or chambers contain just one cell. qPCR is performed using two pairs of primers—to a viral marker gene and to a cell marker gene (the rRNA gene). Then the DNA from the droplets or chambers is retrieved and sequenced. The co-amplification of both viral and bacterial marker genes in one droplet or chamber indicates phage-bacteria association. Statistical analysis of these associations reveals the relations between phages and bacterial species inside the community. The method was verified on a microbial population of termite hindgut. Emulsion-paired isolation-concatenation PCR (epicPCR) [83], initially used for profiling bacterial communities [84], was subsequently modified for linking uncultured phages to bacteria. In this approach, single cells from environmental samples are encapsulated in polyacrylamide beads in emulsion oil. Phusion PCR is performed inside each bead. One of the primers to the target gene (viral) carries an overhang matching the 16SrRNA bacterial gene. Co-localisation of the target gene with the host rRNA gene results in fused products that are later detected with next-generation sequencing. The applicability of the techniques above to crAss-like phages is yet to be tested.

## 9. Morphology and Molecular Structure of crAss-like Phages

An early in-depth analysis (combining sensitive PSI-BLAST and HHPred searches for protein homologies) of two representative *Crassvirales* phages, p-crAssphage (now *Carjivirus communis*, family *Intestiviridae*) and IAS virus (now *Paundivirus hollandii*, family *Steigviridae*), managed to assign functions to many of the hypothetical proteins encoded by both genomes [39]. Both viruses appear to have genes grouped into distinct functional modules: the replication module, the transcription module, and the structural protein module (the latter of which can be further split into capsid and tail protein sub-modules). This functional division agrees with other features of their global genome organisation:An inversion of GC-skew likely separates the genomes into two oppositely oriented replichores (one containing the replication module and the other containing the structural and transcription modules) [50]. These functional modules coincide with transcriptional modules of early (regulatory), middle (replication + optional transcription), and late genes (transcription + structural virion proteins), as discussed below (Figure 2).

While the podovirus-like morphology of *Crassvirales* virions had been predicted based on genome analysis, it was not until the isolation of first-cultured *Crassvirales* that this morphology could be confirmed. *Bacteroides* phages crAss001 [18], DAC15, DAC17 [19], and *Cellulophaga baltica* phage phi14:2 [31] (all *Steigviridae*) all share a very similar morphology with isometric icosahedral capsids 77–88 nm in diameter (depending on the microscopy method) and short non-contractile tails of 36–44 nm decorated with appendages of different sizes. The *Bacteroides xylanisolvens* phage crAss002 (*Jahgtovirus secundus*, family *Intestiviridae*) has a capsid of 77 nm and a shorter tail of 18 nm [20]. A crude faecal filtrate rich in *Crassvirales* (mainly *Intestiviridae* and *Crevaviridae*) contained two types of virions with isometric heads (~76.5 nm) and shorter or longer tails [10]. Members of the family *Suoliviridae*, as well as additional proposed epsilon- and delta-groups of *Crassvirales*, have never been isolated in culture or observed microscopically.

A major advancement in understanding the virion structure and functions of individual structural proteins in *Crassvirales* came from a cryo-EM-based reconstruction of crAss001 virions, allowing for atomic-level modelling of many of the capsid and tail proteins [85]. This analysis revealed a T = 9 icosahedral capsid with an outer diameter of 88 nm. The capsid is mainly composed of the HK97-type fold major capsid protein (MCP) and an auxiliary capsid protein (AUX). The structure of the AUX protein is unique to the *Crassvirales*. In addition to these main subunit types, two types of head fibres, trimeric and dimeric, with recognisably Ig-like domains, are inserted into the central pockets formed by the MCP-AUX hexons. These fibres seem to be narrowly specific to particular taxa within *Crassvirales*, with only a fraction of viral diversity in this order having detectable homologues. At the same time, unrelated proteins may have analogous functions in other species of *Crassvirales*. Both types of fibres can potentially be involved in either host recognition, or perhaps binding to the intestinal mucus, increasing the persistence of phage in the gut and preventing its washout (BAM-model of gut phage persistence) [86]. Such a phenomenon has previously been reported, with hoc+T4 coliphage being capable of binding to the gut mucin.

A dodecameric portal assembly is inserted in one of the five-fold symmetrical capsid vertices, providing an entry point for DNA packaging and an interface for docking the tail barrel assembly. Together with the large terminase gene, primase, and the MCP gene, the portal protein gene forms a set of genes universally conserved across the order *Crassvirales*. Zones around the portal crown domains extending into the capsid matrix contain a lower density of packaged DNA and are believed to be occupied by cargo proteins, which eject from the capsid together with, or ahead of, the phage genome in the early stages of infection. These cargo proteins may include 1–3 large subunits of the vRNAP as well as the large proteins C1 and C2, however only the presence of C1 protein can be confirmed from cryo-EM reconstruction. All cargo proteins are likely to be present in a partially unfolded state, facilitating their passage through the portal and the tail barrel. Protein C1 appears to be stretched out, with parts of its sequence visible inside the capsid (around the portal crown) and the tail barrel. Such partially unfolded C1, as well as parts of other cargo proteins and the leading DNA end, may constitute the central body visible inside the portal channel and the tail barrel.

The tail barrel is composed of five stacked dodecameric rings. Rings 1–3 are composed of dedicated subunit types, whereas rings four and five use identical subunits. Interestingly, different clades of *Crassvirales* contain variable numbers of tail-barrel ring protein genes, implying a different number of rings and therefore tail length in different species. This agrees with EM observations of diverse *Crassvirales*. A hexameric tail muzzle protein forms the tail barrel assembly and represents a structural element highly conserved across the *Crassvirales* [10]. This multi-domain protein contains some elements shared with tail tip proteins from other phages, e.g., a ring-joining domain and a β-propeller domain containing the gate loop are similar to the coliphage T7 nozzle protein. At the same time, two Ig-like domains and, most notably, a domain with a novel type of fold, termed crAss fold, located at the very tip of the muzzle structure, are highly unique to the *Crassvirales*. It is believed that the crAss domain, together with Ig-like domains, might be responsible for the docking of phage particles on the cell surface. Conformational changes evoked by this docking event can trigger the opening of the gate located above, in the proximal part of the muzzle, ejecting the capsid contents.

Two types of protein subunits (THA and THB) are responsible for forming a collar structure around the tail neck. This structure forms a docking hub for the attachment of tail fibre/spike assemblies—flexible structures forming the outer and inner “cages” around the tail barrel. While five different proteins are likely to be involved in this assembly in crAss001, the exact structure could not be resolved using cryo-EM. Tail spike subunits in different *Crassvirales* have been annotated as containing carbohydrate binding/lectin-type domains (e.g., the BACON domain in p-crAssphage and related phages) as well as possible glycosyl-hydrolase domains, phosphatase domains, and CHAP domains [1,18,20,47]. Tail spike proteins with CPS-, EPS-, and LPS-depolymerising activity are a typical feature in phages with podovirus-like morphology [87]. It is therefore likely that the tail fibres/spikes of *Crassvirales* act as receptor binding proteins (RBPs) and are responsible for both the recognition of polysaccharide structures covering host cells, and the digestion of CPS/EPS/LPS polysaccharides in order to gain access to the outer membrane for docking. The genomic region occupied by the tail spike genes is characterised by a striking level of diversity, with different lineages of *Crassvirales* containing completely different numbers and types of genes, often with no detectable homology to their counterparts in crAss001 or p-crAssphage [10,18]. Even within narrower groups in *Crassvirales* (just p-crAssphage —*Carjivirus communis*), tail spike proteins are subjects to rapid evolution, e.g., DGR-dependent diversification [44] and evolution through duplication and HGT of BACON repeats [47]. Longitudinal analysis of p-crAssphage genomes persisting in the human gut revealed rapid evolution of tail spike proteins with evidence of positive selective pressure [44,88].

Overall, the genome and virion structure of *Crassvirales* resemble those of coliphage N4 (the latter being slightly smaller at 72 kb genome and 70 nm capsid). Interestingly, coliphage N4 also includes the rare feature of a high-molecular-weight (360 kDa) vRNAP [89,90].

## 10. *Crassvirales* Life Cycles and Phage-Host Dynamics In Vivo and In Vitro

In addition to their high abundance in many individuals, another characteristic feature of the ecology of the *Crassvirales* is the long-term persistence of the same strains at constant levels in individual human gut phageomes. Human individuals followed for periods of months to a year demonstrated stable carriage of individual-specific viral assemblages, including one or several strains of *Crassvirales* [4,13,88]. This is paralleled by the observed long-term stability and individual specificity of bacterial microbiota, including members of the order Bacteroidales [13,60,90,91], which serve as a substrate for crAss-like phage persistence. The adapted co-existence of *Crassvirales* and their hosts manifests in their ability to co-transmit vertically from mothers to infants [88] and co-engraft during faecal microbiota transplantation (FMT) [38].

The long-term persistence of crAss-like phages has been replicated in both in vitro and in vivo models. Phages crAss001 and crAss002 have both been demonstrated to persist at high levels without noticeable effects on bacterial host culture density in serial daily liquid cultures, chemostat cultures, and, in the case of crAss001, in a gnotobiotic model of mice mono colonised with their bacterial host *B. intestinalis* [18,20]. Moreover, the uncultured crAss-like phage phiHSC05, introduced as a part of the human faecal virome, was able to stably engraft in a gnotobiotic mouse model colonised with an artificial community of human gut bacteria [29]. This long-term co-existence may not necessarily be unique to *Crassvirales*, as many tailed and non-tailed phages demonstrate very similar behaviour of relatively “benign” co-existence with their hosts for many generations in the mammalian gut [13,30,92,93,94]. Theoretical explanatory models of phage-host co-existence are listed in Box 2; however, further experimental evidence is needed to resolve which mechanism or combination thereof best describes *Crassvirales* ecology.

Current evidence for various elements in the *Crassvirales* phage life cycle and their interactions with bacterial hosts is available for a small subset of phages, either isolated in pure culture (crAss001, DAC15, crAss002, phi14:2, and the C4-group of phages infecting *B. uniformis*) [21], or transiently enriched/maintained in host cultures in vivo or in vitro (phiHSC05 in oligo community in mice [29], a 104.9 kb crAss-like phage genome associated with *Prevotella stercorea* in an in vitro culture [23]).

Phages crAss001, DAC15, and DAC17 display a narrow host range, being only able to infect phase variants displaying a particular capsular polysaccharide (CPS) and outer membrane protein (OMP) within a single strain of *B. intestinalis* APC919/174 and *B. thetaiotaomicron* VPI-5482, respectively. In the case of crAss001, only the expression of PVR9, one out of five phase-variable CPS loci available in the host strain, is associated with phage sensitivity [22]. PVR9 expression in *B. intestinalis* was associated with strong phage adsorption, suggesting its role as a phage receptor for crAss001. In a similar manner, phages DAC15 and DAC17 were narrowly specific to a single phase-variable CPS (CPS3), out of eight encoded by *B. thetaiotaomicron* VPI-5482, although some marginal infectivity was also observed for CPS7 and CPS8, as well as for an acapsular mutant [19]. Mixed cultures expressing several different CPS types supported phage replication in vitro and in vivo (i.e., mono-colonised mice), but were unable to show lysis in traditional plaque and spot assays, most likely due to masking of plaques by resistant subpopulations.

In long-term co-cultures of crAss001 and *B. intestinalis* APC919/174, sequestration of part of the population in the resistant phase, and constant reversal to sensitivity might be sufficient for phage persistence. In complex community scenarios, concurrent selective pressures may be responsible for driving this backward switch to sensitivity: the metabolic cost of protective CPS expression, concomitant sensitivity to other phage conferred by a CPS protective against one phage [19,95], and complex pressures and trade-offs associated with the role of CPSs as gut colonisation factors in vivo [96]. A summary of CPSs-phage interactions is presented in Figure 3. crAss002 and phages of the C4 group also demonstrate narrow host specificity (*B. xylanisolvens* APCS1/XY and a group of closely related *B. uniformis* isolates, respectively) [21]. Unlike crAss001, DAC15, and DAC17, these phages are unable to form plaques, likely due to the prevalence of resistant phase variants in populations of their hosts. Indeed, long read sequencing of the *B. xylanisolvens* APCS1/XY genome revealed several phase-variable CPS and OMP loci, which can potentially be implicated in driving reversible phage resistance [20].

As mentioned above, *Crassvirales* tail fibres/spikes are most likely required for adsorption to the CPS target and local depolymerisation of the polysaccharide capsule to gain access to the outer membrane. While the exact mechanisms of virion docking and ejection are still unknown, some insights into this process were provided by a recent structural study of crAss001 [85]. Cargo proteins C1 and C2 are likely to be ejected, forming a tunnel through the cell envelope (C1 has an identifiable transmembrane helix), in a manner analogous to phage T7 protein gp14 [97]. This is likely followed by the ejection of the phage-associated vRNAP subunits in a partially unfolded state to allow passage of these through a relatively narrow (<50 Å) channel inside the tail barrel. Ejection of cargo proteins is likely to be closely followed by ejection of the phage genome, both driven by high pressure achieved through tight packaging of DNA and proteins in the phage capsid.

A study of transcription in phage crAss001 [22] identified three main phases: (1) transcription of early genes, starting in less than 10 min after infection, including a transcriptional regulator, two conserved domain proteins with unknown function, and a TROVE domain protein; (2) transcription of middle genes, initiating between 10 and 30 min after infection, including replication, recombination, and nucleotide metabolism functions; and (3) transcription of late genes, initiating between 30 and 60 min after infection, including a large terminase, all structural proteins, and most cargo proteins. A study of transcription in phi14:2, a crAss-like phage with a longer latent period [98], reported early genes (expressed between 40 and 90 min), corresponding to both early and middle genes in the crAss001 study. At the same time, middle and late genes (structural and cargo proteins), reported by Drobysheva et al. as being expressed between 90 and 190 min, corresponded to the late genes in the crAss001 study. Importantly, the latter study also confirmed the RNA polymerase activity of one of the predicted phi14:2 vRNAP in vitro and demonstrated its role in rifampicin-resistant transcription of early genes (i.e., regulatory and replication modules) in vivo.

Despite some discrepancies between the two studies, it appears that the transcription program of *Crassvirales* phages is broadly similar to that of coliphage N4 [89]. Upon ejection of virion contents into the cytosol, vRNAP drives expression of early and middle genes independently of host transcription machinery. The synthesis of vDNAP and other enzymes encoded by the middle genes allows phage genome replication to begin. Exact mode of replication is unknown and likely to differ between different clades of *Crassvirales*. The crAss001 genome (*Steigviridae*) was shown to have long terminal repeats (DTR), whereas the crAss002 genome (*Intestiviridae*) is predicted to replicate, forming terminally redundant, circularly permuted genome copies. Based on the analysis of the phage-to-host genome copy ratio, genome replication of crAss001 begins between 30 and 60 min post infection, following expression of the middle genes with simultaneous expression of the late genes [22]. Transcription of late genes is likely to be carried out by the host RNAP and is therefore expected to be sensitive to repression with rifampicin [98]. Phage packaging is mediated by large terminases and again may vary between different clades depending on genome structure. Precise mechanisms of capsid assembly and packaging are yet to be established. However, early morphological changes (formation of electron-light spots) in the infected cells of *B. intestinalis* indicate the existence of some scaffolding. Cell lysis appears to be mediated by spanins and lysins encoded among the late genes, which can be subjected to translational regulation by suppression of in-frame stop codons [50]. Determination of the exact burst size of *Crassvirales* has proven difficult [22], with reports ranging from ~20 (observed for crAss001 at MOI = 1) to ~160 (DAC15, MOI = 10) virions per cell.

Box 2Long-term persistence of bacteriophages in the gut ecosystem. The ability of bacteriophages to persist long-term (months and years) in complex microbial communities such as the human gut microbiota alongside the populations of their bacterial hosts is a well-documented phenomenon [13,42]. At least three groups of mechanisms can be invoked to explain this persistence. (i) Spatial segregation of phages and their hosts in the gut and the “source-sink” model: macro- and micro-anatomy of the gut (villi, glands, mucin layer), as well as self-organised bacterial refuges (biofilms), can create areas of bacterial proliferation (“source”) inaccessible to phages. Release of bacterial cells into the gut lumen or liquid phase provides gut phages with a constant supply of bacterial prey (“sink”). This mechanism can enable phage-host co-existence with minimal co-evolution [99,100]. (ii) Ecological population dynamics of strictly lytic phage and its host under conditions of resource limitation/variability: Lotka-Volterra equations and the “kill-the-winner” model predict that predator and prey can co-exist in a planktonic environment at the point of equilibrium or with cyclic fluctuations of their densities. Under the “kill-the-winner” hypothesis, phage-mediated population control together with the competition for a limited resource favours the maintenance of high diversity in a complex community (even in the absence of co-evolution) [101,102]. (iii) Antagonistic co-evolution and the cost of resistance: a number of theoretical frameworks are often employed to describe phage-bacteria co-evolution, such as the “arms race” dynamics (resistance mutations in the host followed by host range counter-mutations in the phage) and “fluctuating selection” dynamics (Red Queen evolution), the latter leading to diversification of bacterial and phage populations [42,101,103,104]. Longitudinal observations show evidence that selective pressure driving evolution of crAss-like phage genes is particularly strong for tail components (including RBPs), indicating an arms race/Red Queen type of dynamics between phages and their hosts [13,46,88].

## 11. Conclusions

The accidental discovery of p-crAssphage, the most abundant human gut bacteriophage, by cross-assembly of metagenomic reads in 2014 revealed how little we knew about the diversity and the role of bacterial viruses in the human gut microbiome and other complex microbial communities. The known viral diversity of dsDNA-tailed phages has bloomed in recent years due to the increasing availability of shotgun metagenomic and metaviromic data, improved virome analysis pipelines, sensitive protein homology search tools, and expanding collections of reference genomes [7,9,105]. The range of known crAss-like phages has expanded from a few “orphan” species into the expansive and highly diverse new viral order of *Crassvirales* [8,46]. Still, the phylogeny of the *Crassvirales* and their evolutionary relationships with other dsDNA viruses remain largely unexplored.

Metagenomics and metaviromics have proven themselves powerful technologies for virus discovery, host prediction, taxonomy, and abundance and distribution analysis [28]. However, in silico methods can be limited in their tangible revelations on the molecular biology, host range, and ecological impacts of crAss-like phages. Emerging engineering advances in single-cell sequencing techniques, viral tagging, and proximity ligation are set to provide new insights into the biology of diverse crAss-like phages.

The first in-depth characterisation of a crAss-like phage, crAss001, while providing a complete genome and data on virion structure and certain aspects of its life cycle [18,22,84], posed numerous new questions about the biology of this phage taxon. Being apparently a strictly lytic phage, crAss001 is capable of long-term persistence in the culture of its host *Bacteroides intestinalis*, which was shown to be associated with dynamic phase variation of capsular polysaccharide types. However, is that the only mechanism responsible? Could alternative mechanisms of persistence (lysogeny, pseudo-lysogeny, the carrier state, or chronic infection “hibernation” inside infected cells) also be part of it? Available evidence does not support the ability for true lysogeny in crAss-like phages, but other mechanisms cannot be ruled out.

Phage crAss002 infects its host *Bacteroides xylanisolvens* without forming plaques, despite being apparently a lytic phage [20]. Is that a result of phase variation greatly shifting towards the insensitive phase? Such phage-host interactions may be puzzling to the observer, but is it uniquely specific to the *Bacteroides*/crAss-like phage systems, or is it a widespread type of dynamic between phages and their bacterial hosts in natural settings, a better winning strategy for the phage? Perhaps the “classical” lytic process, favoured by many studies of phage biology, is a very artificial scenario that can only be achieved in perfect conditions (exactly the right growth phase, MOI, etc.).

As discussed in our other recent review [30], bacteriophages can be seen as an integral component of complex microbial communities, evolutionary partners rather than antagonists of bacteria, responsible for maintenance of taxonomic and functional diversity, stimulating resilience and adaptability of the microbiota. The tandem of *Bacteroides* bacteria and crAss-like phages in the gut might be an epitome of such relationships.

## Figures and Tables

**Figure 1 biomolecules-13-00584-f001:**
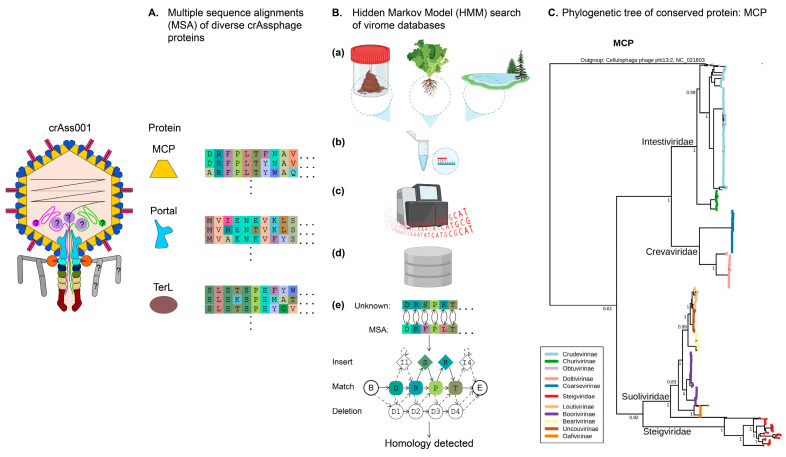
Bioinformatic approaches to discovering and classifying crAss-like phages. (**A**), Multiple sequence alignments (MSA) of the evolutionary conserved phage proteins such as major capsid protein (MCP), portal protein, and terminase large subunit (TerL) contain signatures specific to the order *Crassvirales*. MSAs are converted to profile HMMs (Hidden Markov Models). (**B**), Discovery and classification of new *Crassvirales*. Candidate genomes are collected from varied environments such as stool, soil, or aquatic biomes (**a**); shotgun sequencing and assembly of total or VLP-enriched DNA (**b**,**c**) yields large assembly databases (**d**); translated proteins are compared to MSAs of conserved *Crassvirales* proteins through sensitive position-specific homology detection algorithms (PSI-BLAST, HMM, and HHM). (**C**), Using MCP as a taxonomic marker protein, a maximum likelihood phylogenetic tree can be constructed to infer relatedness between families (main first-order branch names) of crAss-like phages [48].

**Figure 2 biomolecules-13-00584-f002:**
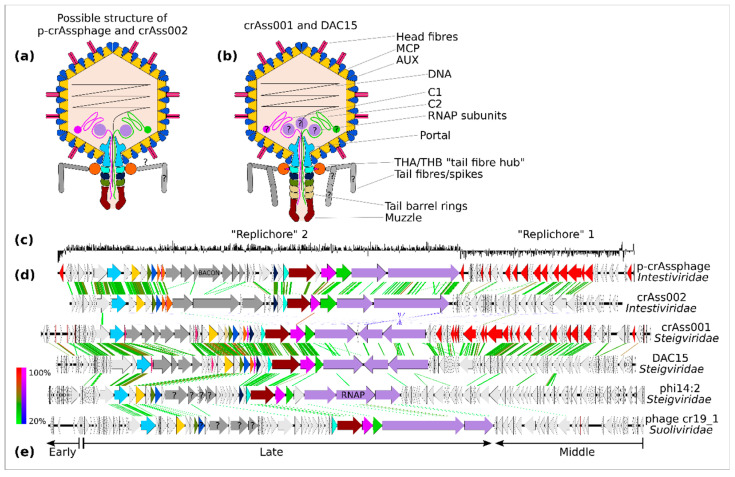
Virion and genome structure of the typical representatives of the order *Crassvirales*. A schematic structure of *Intestiviridae* virions (**a**) is extrapolated from the cryoEM structural data available for *Steigviridae* (**b**). Colours of the protein subunits (**a**,**b**) are consistent with ORF colouring (**d**). GC-skew graph of the p-crAssphage genomes shows clear separation into two possible replichores (**c**). Despite significant divergence in the amino acid sequence of the encoded proteins, all families show clear genome collinearity (**d**). Three transcriptional modules can be identified: early (regulatory), middle (DNA replication), and late (structural proteins and other virion components), as exemplified by a *Suoliviridae* genome (**e**).

**Figure 3 biomolecules-13-00584-f003:**
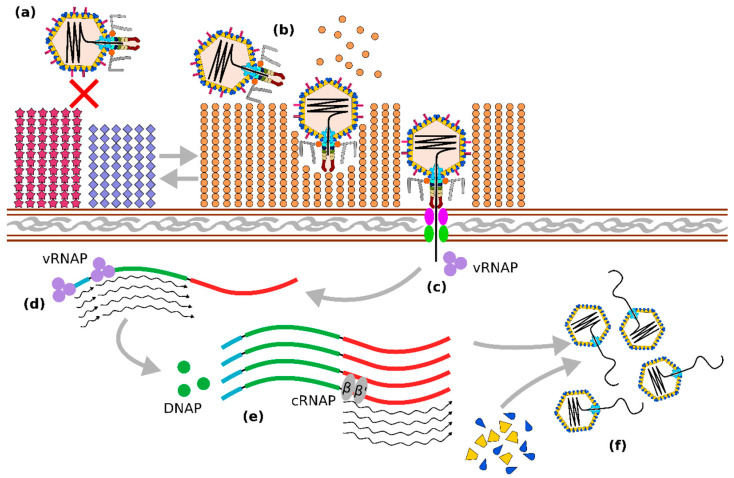
Some known elements of the *Crassvirales* life cycle. Adsorption and infection of *Bacteroides* cells by crAss-like phages are regulated by the expression of alternating phase-variable capsular polysaccharide (CPS) types. Expression of non-permissive CPS prevents phage infection (**a**). while switching to permissive CPS allows phage attachment, hydrolysis of CPS by tail-associated depolymerase enzymes, and ejection of phage contents (**b**). Phage DNA is co-injected with cargo proteins, among which are virion RNA polymerase (vRNAP) subunits responsible for transcription of early (regulatory) and middle (DNA replication) genes (**d**). Formation of DNA polymerase (DNAP) as well as other enzymes, allows for DNA replication through an unknown mechanism, followed by expression of the late genes (large terminase, structural proteins, virion cargo proteins), carried out by the cellular bacterial RNA polymerase complex (cRNAP), (**e**). This results in the formation of new virions and the packaging of phage DNA (**f**).

## Data Availability

Not applicable.

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
