# Peer review of "Bacteriophages of the Order Crassvirales: What Do We Currently Know about This Keystone Component of the Human Gut Virome?"

_biomolecules, 2023, doi:10.3390/biom13040584_

Round 1

Reviewer 1 Report

This review provide a globel view of recently discovered bacteriophage group from order Crassvirales that is prevelent at least in mammalian gut and commonly use Bacteroidetes as their hosts. The fact given by all advancements is that they might be benefical to the maintanance of the health of human or other animals.  This review summarises available information on genomics, diversity, taxonomy, and ecology  of this largely uncultured viral taxon. With experimental data available from a handful of cultured  representatives, the review highlights key properties of virion morphology, infection, gene expression and replication processes, phage host dynamics. It is of greatly value to discover viral tools to manage human or aminal health for wide reseachers in the future.

Author Response

This review provides a global view of recently discovered bacteriophage group from order Crassvirales that is prevalent at least in mammalian gut and commonly use Bacteroidetes as their hosts. The fact given by all advancements is that they might be beneficial to the maintenance of the health of human or other animals. This review summarises available information on genomics, diversity, taxonomy, and ecology of this largely uncultured viral taxon. With experimental data available from a handful of cultured representatives, the review highlights key properties of virion morphology, infection, gene expression and replication processes, phage host dynamics. It is of greatly value to discover viral tools to manage human or animal health for wide researchers in the future.

We thank the reviewer for the positive score and comments.

Reviewer 2 Report

Smith et al. offer a review of the current state of the art in the research of Crassvirales, one of the major components of the human gut virome. As expected from a group of active microbiome scholars, the review is generally well-written and informative; it summarizes the emerging picture of morphology, physiology and molecular biology of crass-like phages.

Minor comments:

CRISPR-targeted crassphage discovery – there is a study worth mentioning, “Comprehensive discovery of CRISPR-targeted terminally redundant sequences in the human gut metagenome: Viruses, plasmids, and more.” PMID: 34673779

Bayfield et al., “Structural atlas of the most abundant human gut virus,” In Review, preprint, Aug. 2022. – is the permanent reference available yet?

A typo in Line 159: “other mammalian faces such as rhesus macaque, pig and cat <…>”

Author Response

Smith et al. offer a review of the current state of the art in the research of Crassvirales, one of the major components of the human gut virome. As expected from a group of active microbiome scholars, the review is generally well-written and informative; it summarizes the emerging picture of morphology, physiology and molecular biology of crass-like phages.

We thank the reviewer for this positive appraisal of our work!

Minor comments:

CRISPR-targeted crassphage discovery – there is a study worth mentioning, “Comprehensive discovery of CRISPR-targeted terminally redundant sequences in the human gut metagenome: Viruses, plasmids, and more.” PMID: 34673779

Reference to Sugimoto et al., 2021 added to line 400, 405, box. 1.

Bayfield et al., “Structural atlas of the most abundant human gut virus,” In Review, preprint, Aug. 2022. – is the permanent reference available yet?

Bayfiled et al. Is unfortunately still available only as a pre-print.

A typo in Line 159: “other mammalian faces such as rhesus macaque, pig and cat <…>”

Typo corrected.

Reviewer 3 Report

This is a well discussed review on a highly talked about virome, the crAssphage. The ample of information gathered here will be useful for researchers/readers interested on this specific phage. I would like to thank the authors for their rigorous discussion about its structure and dynamics in human gut. 

In the review, I found 7 self-citations, and I think it is not quite acceptable. In this connection, I would like to ask the authors to reduce the self-citations and cite other manuscripts instead.

Author Response

This is a well discussed review on a highly talked about virome, the crAssphage. The ample of information gathered here will be useful for researchers/readers interested on this specific phage. I would like to thank the authors for their rigorous discussion about its structure and dynamics in human gut.

We thank the reviewer for this positive appraisal of our work!

In the review, I found 7 self-citations, and I think it is not quite acceptable. In this connection, I would like to ask the authors to reduce the self-citations and cite other manuscripts instead.

We thank the reviewer noting this and, in other circumstances, would agree with this criticism. There is, however, a very limited number of groups in the world that focus their work on Crassvirales, the APC Gut Phageomics Lab led by Colin Hill and Andrey Shkoporov being one of them. In this context, several self-citations are unavoidable and to be expected. We in good conscience cited papers that are relevant and informative of the current state of the art on crAssphage research. The elimination of any of the cited papers would come to the detriment of the content of the review.

Reviewer 4 Report

The manuscript presents an authoritative review of the biology and history of crAssphages. It is well written and informative, and overall does a great job at synergizing knowledge on the biology of this group. I have only two minor points of criticism:

1) I’d like to see some clarification on what is meant by “crAss-like” phages – members of the Crassvirales outside of the Intestiniviridae/Alpha/p-crAssphage clade? Or do the authors just mean all crassphages at this point, with the term crass-like a historical remnant from before they were unified in the Crassvirales? In that case, wouldn’t it be better to just change crAss-like phages to crassphages?

2) A 2017 phylogeny by Koonin & Yutin using the crassphage structural gene module (https://doi.org/10.1016/j.tim.2020.01.010) resolved phi14:2, isolated in 2013 by Holmfeldt et al., as a close relative of IAS virus. As far as I can reconstruct, this was the first time it was noticed in a publication that a crassphage had been isolated already – a year before Dulith’s first crassphage paper in 2014 and Shkoporov’s isolation of crAss001 in 2018, which only mentions the uncultured IAS virus as a close relative of crAss001 but omits phi14:2. The current ICTV taxonomy places all three phages into the family Steigviridae and subfamily Asinivirinae – as mentioned in the text. As such, I don’t think the review should call crAss001 the first crAss-like representative (for example line 639) and I’d like to see a bit more emphasis on how phi14:2 was isolated first, although the significance of it might not have been realized at that point.

Minor line comments:

Line 18: should say bacterial viruses with DNA genomes?

Line 76: Cite Holmfeldt et al. 2013 (also note that the isolation of this crassphage took place before the Dulith paper)

Line 141: The  relevant literature on the isolation of these phages should be cited here

Line 187: “Primordial” implies a bit too far back in biological history…

Line 200: Not sure what is meant with “creative” here

Line 203 : It’s unusual to define dN/dS as “nucleotide polymorphisms and substitution rates” – it’s rate of non-synonymous to synonymous changes.

Line 229: Again, phi14:2 is omitted

Line 236: I believe figure 1C is identical to the ICTV proposal (ref 46) and should be cited here

Line 226: Toggle implies both polymerases are encoded in the genome – maybe a different word could be used

Line 337: I think this is supposed to say “thought to be”

Line 606-607: crAss001 is here listed as both Steigviridae and Intestiniviridae – I believe there’s a mixup

Author Response

The manuscript presents an authoritative review of the biology and history of crAssphages. It is well written and informative, and overall does a great job at synergizing knowledge on the biology of this group. I have only two minor points of criticism:

We thank the reviewer for this positive appraisal of our work!

1) I’d like to see some clarification on what is meant by “crAss-like” phages – members of the Crassvirales outside of the Intestiniviridae/Alpha/p-crAssphage clade? Or do the authors just mean all crassphages at this point, with the term crass-like a historical remnant from before they were unified in the Crassvirales? In that case, wouldn’t it be better to just change crAss-like phages to crassphages?

The term “crAssphage” is used inconsistently in the literature and it some contexts mean either a specific phage “crAssphage” described by Dutilh et al. (2014) or the whole taxonomic group Crassvirales. Because of that, we prefer to refer to the original crAssphage as “prototypical”, or p-crAssphage, whereas the trivial non-taxonomic term “crAss-like phages” (coined by Yutin et al. in their 2017 article) is used to refer to diverse phages in the order Crassvirales. A note in line 33 was added to draw attention to this naming inconsistency.

2) A 2017 phylogeny by Koonin & Yutin using the crassphage structural gene module (https://doi.org/10.1016/j.tim.2020.01.010) resolved phi14:2, isolated in 2013 by Holmfeldt et al., as a close relative of IAS virus. As far as I can reconstruct, this was the first time it was noticed in a publication that a crassphage had been isolated already – a year before Dulith’s first crassphage paper in 2014 and Shkoporov’s isolation of crAss001 in 2018, which only mentions the uncultured IAS virus as a close relative of crAss001 but omits phi14:2. The current ICTV taxonomy places all three phages into the family Steigviridae and subfamily Asinivirinae – as mentioned in the text. As such, I don’t think the review should call crAss001 the first crAss-like representative (for example line 639) and I’d like to see a bit more emphasis on how phi14:2 was isolated first, although the significance of it might not have been realized at that point.

The reviewer is absolutely correct here. Sentence added to highlight the isolation of phi14:2 before the discovery of the prototypical crAssphage, line 365.

Minor line comments:

Line 18: should say bacterial viruses with DNA genomes?

Wording changed to “double-stranded DNA bacteriophages”.

Line 76: Cite Holmfeldt et al. 2013 (also note that the isolation of this crassphage took place before the Dulith paper)

Citation to Holmfeldt et al. 2013 added.

Line 141: The relevant literature on the isolation of these phages should be cited here

Citation added: Yutin et al., 2018.

Line 187: “Primordial” implies a bit too far back in biological history…

Term changed to “long-standing”.

Line 200: Not sure what is meant with “creative” here

Term changed to “versatile”.

Line 203: It’s unusual to define dN/dS as “nucleotide polymorphisms and substitution rates” – it’s rate of non-synonymous to synonymous changes.

Sentence about single nucleotide polymorphism corrected.

Line 229: Again, phi14:2 is omitted.

Holmfeldt et al., 2013 citation added.

Line 236: I believe figure 1C is identical to the ICTV proposal (ref 46) and should be cited here

Citation added to figure description.

Line 226: Toggle implies both polymerases are encoded in the genome – maybe a different word could be used.

Line changed to clarify either or polymerase gene is carried in the genomes of Zeta and Beta groups.

Line 337: I think this is supposed to say “thought to be”

Missing word added.

Line 606-607: crAss001 is here listed as both Steigviridae and Intestiniviridae – I believe there’s a mixup

Typo corrected to read crAss002.